# Mechanical Behavior of Human Cancellous Bone in Alveolar Bone under Uniaxial Compression and Creep Tests

**DOI:** 10.3390/ma15175912

**Published:** 2022-08-26

**Authors:** Bin Wu, Yang Wu, Mao Liu, Jingjing Liu, Di Jiang, Songyun Ma, Bin Yan, Yi Lu

**Affiliations:** 1College of Mechanical and Electronic Engineering, Nanjing Forestry University, Nanjing 210037, China; 2Department of Orthodontics, The Affiliated Stomatological Hospital of Nanjing Medical University, Nanjing 210029, China; 3Jiangsu Province Key Laboratory of Oral Diseases, Nanjing 210029, China; 4Jiangsu Province Engineering Research Center of Stomatological Translational Medicine, Nanjing 210029, China; 5Institute of General Mechanics, RWTH-Aachen University, 52062 Aachen, Germany

**Keywords:** alveolar bone, cancellous bone, compress test, hyperelastic model, viscoelastic model

## Abstract

In the process of orthodontic treatment, the remodeling of cancellous bone in alveolar bone (in this paper, cancellous bone in alveolar bone is abbreviated as CBAB) is key to promoting tooth movement. Studying the mechanical behavior of CBAB is helpful to predict the displacement of teeth and achieve the best effect of orthodontic treatment. Three CBAB samples were cut from alveolar bone around the root apex of human teeth. A uniaxial compression test was used to study the transient elastic properties of CBAB. A creep test was used to study the time-dependent viscoelastic properties of CBAB. Both tests were carried out at the loading rates of 0.02 mm/min, 0.1 mm/min and 0.5 mm/min. The results revealed that CBAB is a nonlinear viscoelastic and hyperelastic material. The stress–strain curve obtained from the uniaxial compression test could be divided into three stages: the collapse stage of the front section, the exponential stage of the middle section and the almost linear stage of the rear end. According to the strain–time curve obtained from the compression creep test, a trend of increasing strain over time was relatively obvious within the first 30 s. After 200 s, the curve gradually tended to plateau. Four hyperelastic models and three viscoelastic models were used to fit the test data. Finally, the fifth-order polynomial hyperelastic model (coefficient of determination “R^2^ > 0.999”) was used to describe the hyperelastic properties of CBAB, and the seven-parameter model of the generalized Kelvin modified model (“R^2^ > 0.98”) was used to describe the viscoelastic properties of CBAB.

## 1. Introduction

Alveolar bone comprises indigenous alveolar bone, cortical bone and cancellous bone. Cancellous bone, located between cortical bone and indigenous alveolar bone, is the where the majority of bone remodeling occurs. In the process of orthodontic treatment, the force is conducted through the periodontal ligament to the alveolar bone. Subsequently, bone resorption happens at the compressive site, enabling the tooth to be moved. Alveolar bone remodeling occurs during bone resorption, and cancellous bone is the main site of bone remodeling [1,2]. Previous studies have shown that [3,4,5,6] the direction and speed of tooth movement are mainly related to the stress distribution in CBAB under stress. Therefore, in order to quantify the relationship between orthodontic force and tooth movement, it is necessary to further clarify the stress response state of CBAB under stress so as to help improve the clinical treatment effect [7].

Many scholars have conducted mechanical tests on cancellous bone in animal or human vertebrae, femur and other parts, and they have constructed corresponding material mechanics models. Hosseinzadeh [8] et al. established the hyperelastic model of demineralized and deproteinized bovine cortical femur bone. The accuracy of this model was verified through an uniaxial compression test. They found that the Mooney–Rivlin and the Ogden models could not accurately predict the mechanical responses of demineralized and deproteinized bovine cortical femur bone, while the general exponential–exponential and general exponential–power law models have good agreement with their experimental results. Pawlikowski [9] et al. proposed a new constitutive model of human trabecular bone. The elastic response was described with the hyperelastic Mooney–Rivlin model, while the viscoelastic effects were considered via the means of the hereditary integral, in which stress depends on both time and strain. The accuracy of this model has been verified by stress relaxation and indentation tests. Li [10] et al. conducted compression tests on the cervical cancellous bone of piglets (child surrogates) from different directions at strain rates of 0.01, 0.1, 1 and 10/s, and developed a strain-rate-dependent transverse isotropic elastic–plastic constitutive model to describe vertebral behavior. In order to reflect the anisotropic characteristics of cancellous bone, Megías [11] et al. presented a new model for the estimation of the elastic properties of lamellar tissue, which included the bone mineral density and the microporosity. The model addressed the numerical modeling of cancellous bone damage using an orthotropic failure criterion and a discrete damage mechanics analysis, including a novel approach for the tissue elastic properties aforementioned. Although some progress has been made in the research on the mechanical properties and constitutive modeling of cancellous bone, there are great differences between animal cancellous bone and human CBAB. The constitutive models established by previous scholars have also varied and can hardly be applied on human CBAB. Therefore, it is necessary to deduce a constitutive model based on human CBAB.

In this study, the mechanical properties of CBAB were studied through a uniaxial compression test and compression creep test. Two new models (viscoelastic and hyperelastic model) with high accuracy were built to describe the mechanical properties of CBAB. The mechanical models established in this study will help to improve the modeling accuracy and finite element simulation accuracy, and provide a reference for predicting tooth displacement under orthodontic forces.

## 2. Methods

### 2.1. Sample Preparation

This study was reviewed and approved (No. (2020)234) by the Institutional Review Board of Nanjing Medical University. Maxillary jaw segments were taken from one fresh corpse (male, 40 years old, in good dental health, without any periodontal disease), brought to the laboratory and stored in a refrigerated container at −20 °C for subsequent sectioning. After the periodontal soft tissues were removed from the maxilla, a block containing two complete molars with the surrounding periodontal ligament and alveolar bone was obtained. The preparation of the samples was divided into three steps: First, the crown was removed. Second, the root apex was sliced perpendicular to the long axis of the tooth using a low speed cutting machine (Isomet, Buehler, Lake Bluff, IL, USA). The root apex slices were cut into long strips of equal width. Third, from these strips, the CBAB samples were cut into cubes with an ideal size of about 1.45 × 2 × 2 mm using a vulgar cutting machine. The preparation process of CBAB samples is shown in Figure 1. (Readers can watch the video of sample preparation process in the Appendix A to understand this process more intuitively).

A total of 13 samples were cut and put into test tubes. Normal saline was injected to moisturize the samples, and the test tubes were stored in a freezer at −20 °C. Before testing, the samples were taken out and thawed in a water bath. Three samples, of approximately the ideal size, were selected. The shape parameters are listed in Table 1.

### 2.2. Mechanical Testing

In this work, a high-accuracy double-column desktop electronic universal material testing machine, Instron 3365, Instron, No. 819, Nanjing West Road, Shanghai, 200041, China was used. Since the samples were small and brittle, the compressing force was limited to 100 N (with a tolerance of 0.1 N). A load cell of 100 N (with a tolerance of 0.1 N) was used, and the strain was calculated through the displacement of the grippers and the initial length. The test setup is shown in Figure 2.

Cancellous bone exhibits both elastic and viscous mechanical properties [12,13,14,15]. The tests were divided into two parts: a uniaxial compression test and a compression creep test. (Readers can watch the video of two tests in the Appendix A to understand the test process more intuitively). The uniaxial compression test was used to study the elastic properties of cancellous bone under instantaneous stress, and the compression creep test was used to study its viscosity characteristics under long-term stress. The uniaxial compression test parameters are listed in Table 2, and the compression creep test parameters are listed in Table 3.

### 2.3. Hyperelastic Constitutive Formula

The hyperelastic model refers to the mechanical modeling of porous media in which there is an elastic potential energy function, which is often used to analyze porous structural materials [16,17,18]. CBAB is constructed from trabecular bone and tissue fluid and can be considered as a porous structural material. Micro-CT imaging from multiple angles has shown that CBAB comprises many pores and irregularly arranged trabecular bone (Figure 3a–c). Figure 3d shows a reconstructed map of CBAB. CBAB’s stress response state is affected by transient forces during orthodontic treatment. Therefore, its elastic properties can be described by the hyperelastic model.

The Mooney–Rivlin model and the Ogden model have been widely used in bone tissue structure by previous scholars, but they cannot predict the stress response of biomaterials with few terms. Darijani [19] et al. presented strain energy density functions based on the power law, exponential, polynomial and logarithmic functions. Their results showed that these strain energy functions can fit the test data of porous structural materials. After this study, Mansouri [20] et al. presented more complete strain energy density functions, such as the general power–law model and the general exponential model. These strain energy functions can predict the mechanical response of bone tissue structure under deformations from different loading conditions. In this paper, a new strain energy function, which can better describe the mechanical response of CBAB under uniaxial compression, is proposed:(1)$$W_{}=\sum_{k=1}^{N}A_{k}(\lambda_{1}^{\alpha_{k}}+\lambda_{2}^{\alpha_{k}}+\lambda_{3}^{\alpha_{k}}-3)$$
where *W* is the strain energy function; *A_k_* is the material parameter; and *λ*_1_, *λ*_2_, *λ*_3_ are the main elongations.

The quasi-static uniaxial compression of CBAB is shown in Figure 4.

Assuming that the material bears the force of the unidirectional load in the main direction 3 and only elastic deformation is considered, the deformation formula of the principal elongation of a material’s hyperelasticity is as follows:(2)$$\{\begin{array}{c} \lambda_{1}=\lambda_{2}\\ \lambda_{3}=\lambda \\ J=\lambda_{1}^{2}\lambda \end{array}$$

The Cauchy stress in principal directions 1 and 2 is:(3)$$\sigma_{1}=\sigma_{2}=0$$

The relationship between Cauchy stress and principal elongation is as follows:(4)$$\sigma_{i}=J^{-1}\lambda_{i}\frac{\partial W_{}}{\partial \lambda_{i}}\;\;\;i=1,\;2,\;3$$

Finally, Formula (1) is introduced into the relationship between the main compressive stress and the main elongation as follows:(5)$$\sigma_{3}=J^{-1}\lambda_{3}\sum_{k=1}^{N}A_{k}(\alpha_{k}\lambda_{3}^{\alpha_{k}-1}\frac{\partial J}{\partial \lambda_{3}})$$

Through derivation of the polynomial function, the relationship between stress and strain was concluded as follows:(6)$$\sigma =\sum_{K=0}^{N}a_{k}\varepsilon^{k}$$
where *α_k_* is the parameter to be identified.

When combined with the trends of the stress–strain curves obtained from the test, the hyperelastic models with quadratic, cubic, forth-order and fifth-order polynomial functions were proposed to describe the uniaxial compression mechanical properties of CBAB. The constitutive equations are as follows:(7)$$\mathrm{Parabola}\;\mathrm{function}——\sigma =a_{0}+a_{1}\varepsilon^{1}+a_{2}\varepsilon^{2}$$
(8)$$\mathrm{Cubic}\;\mathrm{function}——\sigma =a_{0}+a_{1}\varepsilon^{1}+a_{2}\varepsilon^{2}+a_{3}\varepsilon^{3}$$
(9)$$\mathrm{Poly}4\;\mathrm{function}——\sigma =a_{0}+a_{1}\varepsilon^{1}+a_{2}\varepsilon^{2}+a_{3}\varepsilon^{3}$$
(10)$$\mathrm{Poly}5\;\mathrm{function}——\sigma =a_{0}+a_{1}\varepsilon^{1}+a_{2}\varepsilon^{2}+a_{3}\varepsilon^{3}+a_{4}\varepsilon^{4}+a_{5}\varepsilon^{5}$$
where *α*_0_, *α*_1_, *α*_2_, *α*_3_, *α*_4_, *α*_5_ is the parameter to be identified.

### 2.4. Viscoelastic Constitutive Formula

At present, the constitutive model of viscoelasticity is a mathematical expression constructed by various combinations of spring (solid) and viscous pot (fluid) models. Viscoelasticity is a mechanical property of materials that hovers between an elastic solid and a Newtonian fluid. Therefore, when studying the biomechanical properties of CBAB, the superposition of the elastic and viscous properties is regarded as the viscoelastic property of those materials.

CBAB is mainly composed of trabecular bone and tissue fluid, which can be divided into free water and bound water, as shown in Figure 5. Bound water is mainly composed of bone minerals and organic matrix, while free water is pore water distributed within cavities and tubules. Bone minerals provide stiffness and strength, while collagen provides ductility and the crucial ability to absorb energy before fracturing [21]. Therefore, bound water shows the viscoelastic characteristics of general liquid, and trabecular bone reflects the elastic characteristics [22,23]. To describe this unique structure, a mechanical model should be used to describe both the liquid and solid. To fulfill this need, a basic model (three-parameter model) was built by connecting a spring and a Kelvin model in series. The basic model is shown in Figure 6.

The constitutive equation formula is:(11)$$\sigma +\frac{\eta_{1}}{E_{1}+E_{2}}\dot{\sigma }=\frac{E_{1}E_{2}}{E_{1}+E_{2}}\varepsilon +\frac{E_{2}\eta_{2}}{E_{1}+E_{2}}\dot{\varepsilon }$$

Through the Laplace transformation and inverse transformation of Formula (11), the creep compliance formula is derived as follows:(12)$$J(t)=\frac{\varepsilon (t)}{\sigma_{0}}=\frac{1}{E_{2}}+\frac{1}{E_{1}}(1-e^{-\frac{E_{1}}{\eta_{1}}t})=\frac{E_{1}+E_{2}}{E_{1}E_{2}}-\frac{1}{E_{1}}e^{-\frac{E_{1}}{\eta_{1}}t}$$

To make the generalized Kelvin model more consistent with the creep test phenomenon of CBAB, the five-parameter and seven-parameter models of the generalized Kelvin modified model are proposed in Figure 7.

As shown in Figure 7, the sticking pot of one Kelvin model was removed, and then two Kelvin models were connected in series. According to the above formula, the creep compliance formula of the five-parameter Kelvin modified model can be deduced as:(13)$$J(t)=\frac{\varepsilon (t)}{\sigma_{0}}=\frac{1}{E_{0}}+\frac{1}{E_{1}}(1-e^{-\frac{E_{1}}{\eta_{1}}t})+\frac{1}{E_{2}}(1-e^{-\frac{E_{2}}{\eta_{2}}t})$$

Similarly, the creep compliance formula of the seven-parameter model is derived as:(14)$$J(t)=\frac{\varepsilon (t)}{\sigma_{0}}=\frac{1}{E_{0}}+\frac{1}{E_{1}}(1-e^{-\frac{E_{1}}{\eta_{1}}t})+\frac{1}{E_{2}}(1-e^{-\frac{E_{2}}{\eta_{2}}t})+\frac{1}{E_{3}}(1-e^{-\frac{E_{3}}{\eta_{3}}t})$$

The three-parameter, five-parameter and seven-parameter models of the generalized Kelvin modified model were built to describe the viscoelastic properties of CBAB.

## 3. Result

### 3.1. Uniaxial Compression

In the uniaxial compression test, normal saline was added to moisturize the samples, and the interval between the two tests was one hour. The stress–strain curve obtained from the test is shown in Figure 8. This test studied the transient elastic properties of CBAB.

The curve can be roughly divided into three stages: The first stage is the collapse stage, where stress changes little with strain. The second stage is a curve similar to an exponential relationship. The third stage is an almost a linear curve.

From the test results, we can draw the following three conclusions: First, CBAB is not a simple linear elastic feature but a non-linear elastic feature. Second, when a sample is compressed to a strain of 0.06, the higher the compression rate, the greater the stress. Third, the curves of the three samples at the same rate are roughly similar, but their elastic moduli differ due to differences in their internal structure and the arrangement of trabecular bone.

### 3.2. Compression Creep

The compression creep test was carried out at room temperature (25 °C). This test studied the time-dependent viscoelastic properties of CBAB.

The strain–time curve obtained from the compression creep test is shown in Figure 9 and Figure 10. During the first 30 s, the trend of strain increasing with time was relatively obvious (Figure 9), and the curve gradually plateaued after 200 s (Figure 10).

The creep function of viscoelastic materials is usually expressed by creep compliance *J(t)*.

The creep compliance–time curve was calculated according to the creep compliance formula *J(t)* = ε(*t*)/б_0_, as shown in Figure 11.

From the creep compliance–time curve in Figure 11, it can be seen that (1) initially, creep compliance increased rapidly with time and then plateaued after 200 s; (2) the larger the loading rate was, the smaller the creep compliance was before load holding.

In the creep state, the strain and creep compliance of CBAB changed little with time, which is consistent with the characteristics of general viscoelastic materials.

## 4. Discussion

The mechanical analysis of the stress–strain curve and the creep compliance–time curve showed that CBAB is a nonlinear hyperelastic and viscoelastic material. To accurately describe its mechanical properties, four hyperelastic and three viscoelastic models were proposed. The models were fitted to the test data to determine the parameters of the constitutive model. The coefficient of determination *R*^2^ was used to evaluate the accuracy of the fitting. *R*^2^ varies between 0 to 1 such that an *R*^2^ value closer to 1 indicates a better fit.

Figure 12 displays the fitting curves of four hyperelastic models: Parabola, Cubic, Poly4 and Poly5. The least squares method was used for the non-linear fitting of uniaxial compression test data. The coefficients of determination *R*^2^ values were integrated in the form of a bar chart, as shown in Figure 13.

According to the fitting curves and histograms, the fitting degree of the four models at the rates of 0.02 mm/min and 0.1 mm/min were higher than 0.99 (Figure 13), indicating that all the models can describe the mechanical properties of CBAB at a relatively low loading rate.

However, at the rate of 0.5 mm/min, there is an obvious deviation between the Parabola model and the test data (Figure 12c,f,i). In addition, it can be seen that there is a large difference between the *R*^2^ values of the Parabola model and the other models at the rate of 0.5 mm/min (Figure 13). The Poly5 model curve passes through the most test data points (Figure 12), and the fitting degree coefficient is the largest, reaching higher than 0.999. Therefore, the fifth-order polynomial hyperelastic model can be used to describe the elastic properties of CBAB.

Next, the three-parameter, five-parameter and seven-parameter models of the generalized Kelvin modified model were fitted to the creep compliance–time curves obtained from the compression creep test. The fitting curves are shown in Figure 14. The coefficients of determination *R*^2^ values were integrated in the form of bar chart, as shown in Figure 15.

According to the fitting curves and histograms, there is a certain deviation between the three-parameter model, the five-parameter model and the test data (Figure 15). The seven-parameter model fits well with the test data, and the degree of fitting is higher than 0.98 (Figure 15). Therefore, the seven-parameter model can be used to describe the viscoelastic mechanical characteristics of CBAB.

## 5. Conclusions

In this study, the mechanical properties of CBAB were studied experimentally, and a new constitutive model of CBAB was proposed based on the test data.

In the uniaxial compression test, CBAB behaved as a non-linear hyperelastic and viscoelastic material. The stress–strain curves can be divided into three stages: the collapse stage of the front section, the exponential stage of the middle section and the almost linear stage of the rear end. In the compression creep test, the strain increased during the first 30 s and then gradually began to plateau after 200 s.

To accurately describe the mechanical properties of CBAB, four different hyperelastic and viscoelastic models were proposed and evaluated by fitting them to the test data. The results show that the fifth-order polynomial hyperelastic model can describe the instantaneous elastic characteristics of CBAB “(R^2^ > 0.999)”, while the seven-parameter model of the generalized Kelvin modified model can describe the time-dependent viscoelasticity of CBAB “(R^2^ > 0.98)”.

## Figures and Tables

**Figure 1 materials-15-05912-f001:**
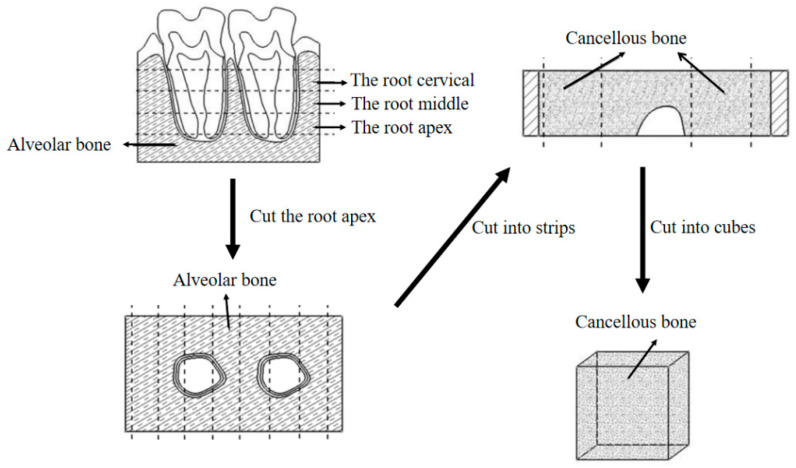
Flow chart of CBAB preparation.

**Figure 2 materials-15-05912-f002:**
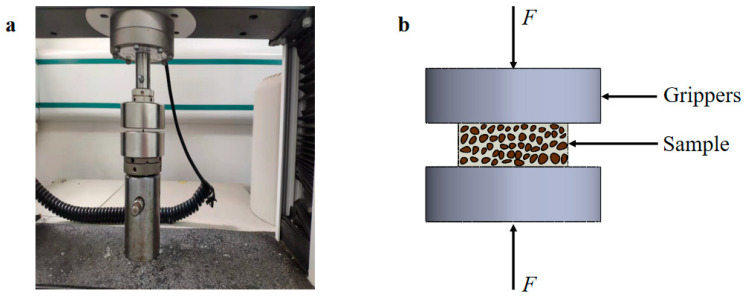
(**a**) Photo of compression fixture, (**b**) Schematic diagram of sample stress.

**Figure 3 materials-15-05912-f003:**
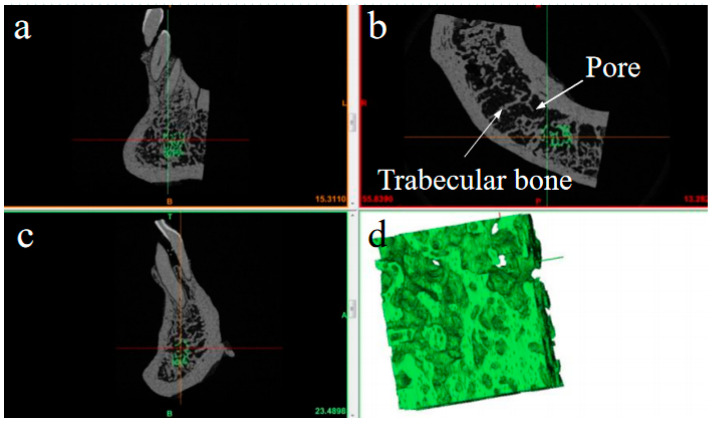
Structure of CBAB under micro CT. (**a**–**c**) Micrographs of periodontal tissue from different perspectives; (**d**) the reconstruction map of CBAB.

**Figure 4 materials-15-05912-f004:**
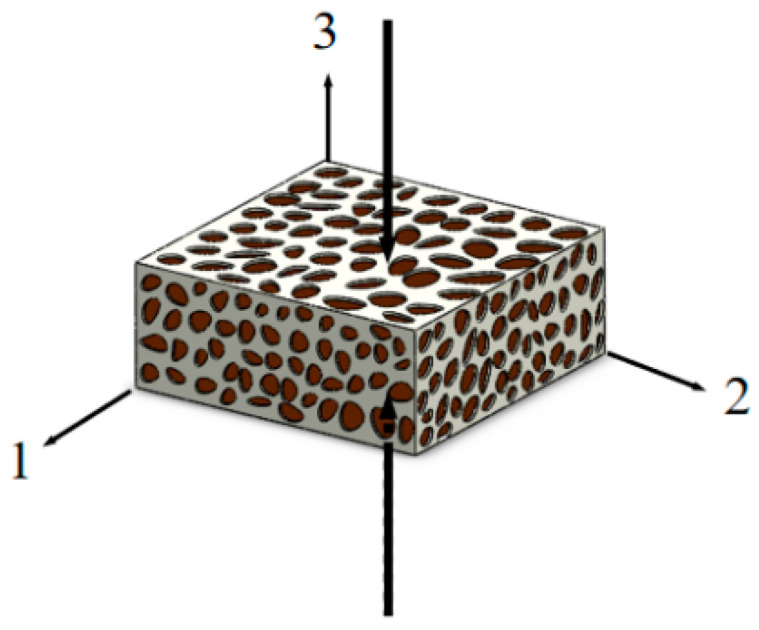
Stress diagram.

**Figure 5 materials-15-05912-f005:**
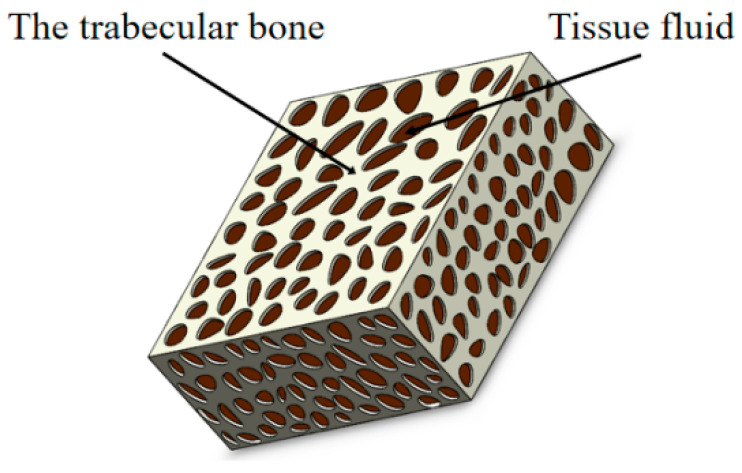
CBAB model.

**Figure 6 materials-15-05912-f006:**
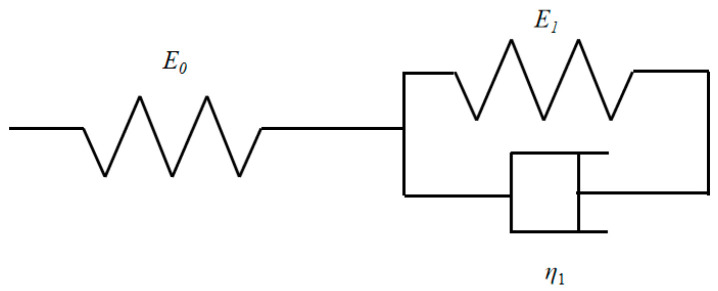
The basic model.

**Figure 7 materials-15-05912-f007:**
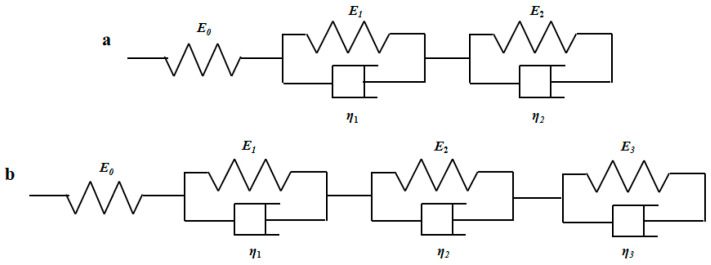
(**a**) Five-parameter Model, (**b**) Seven-parameter model.

**Figure 8 materials-15-05912-f008:**
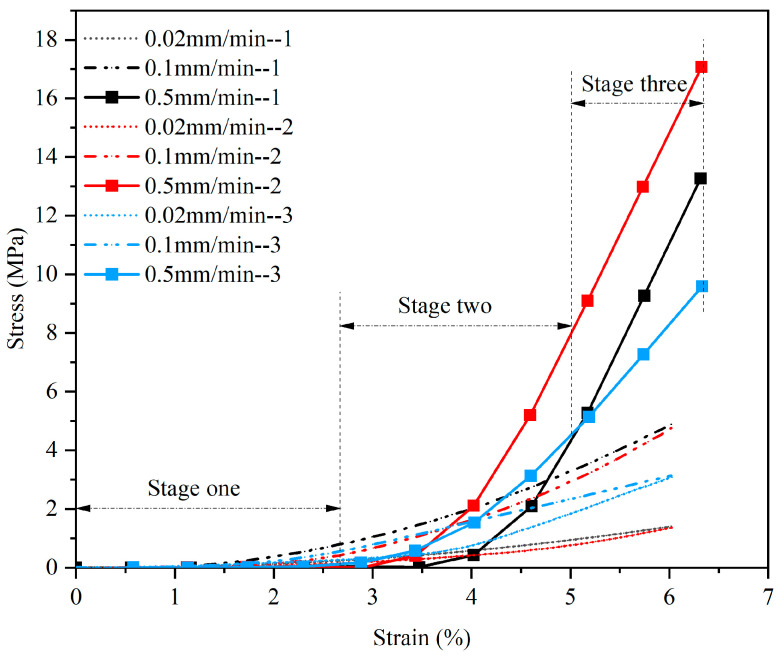
Stress–strain curves of uniaxial compression test.

**Figure 9 materials-15-05912-f009:**
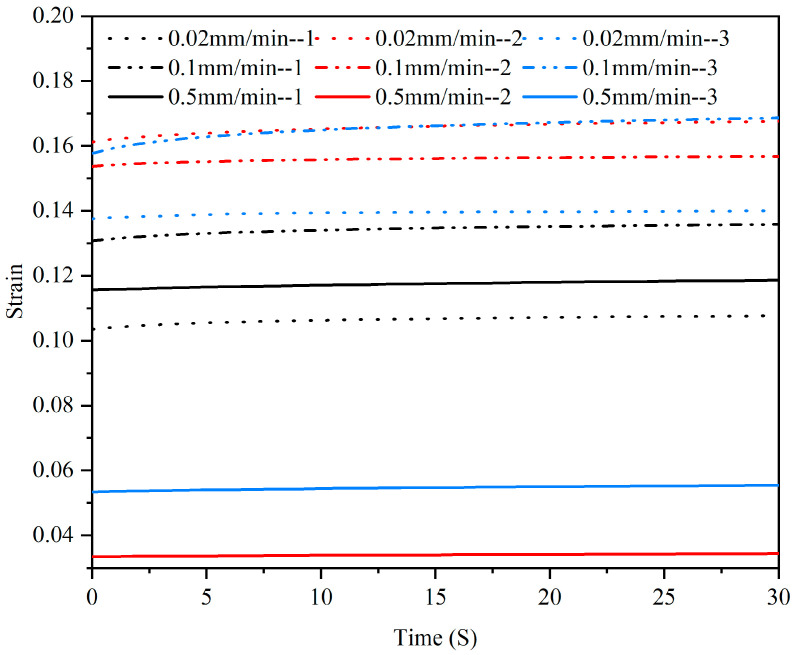
Creep strain–time curves of three samples at different rates in the first 30 s.

**Figure 10 materials-15-05912-f010:**
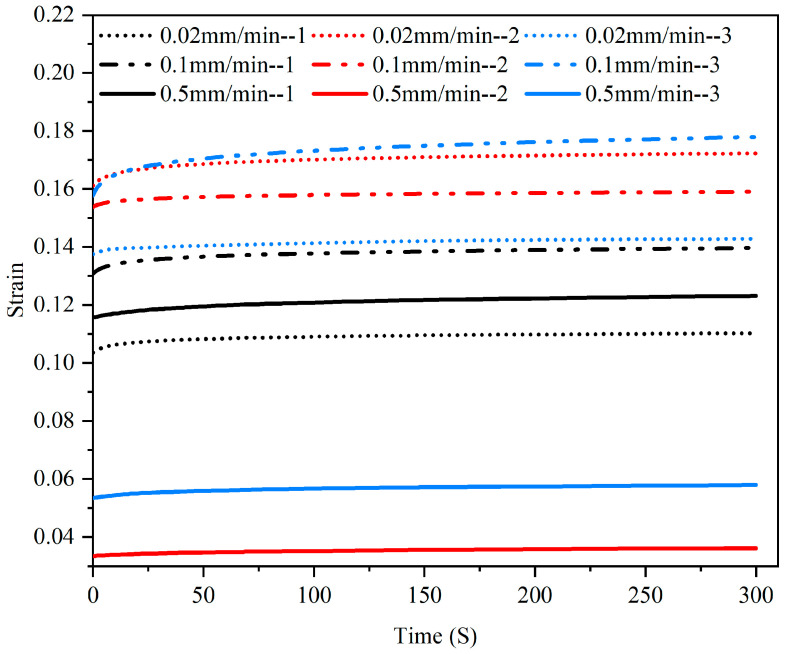
Creep strain–time curves of three samples at different rates.

**Figure 11 materials-15-05912-f011:**
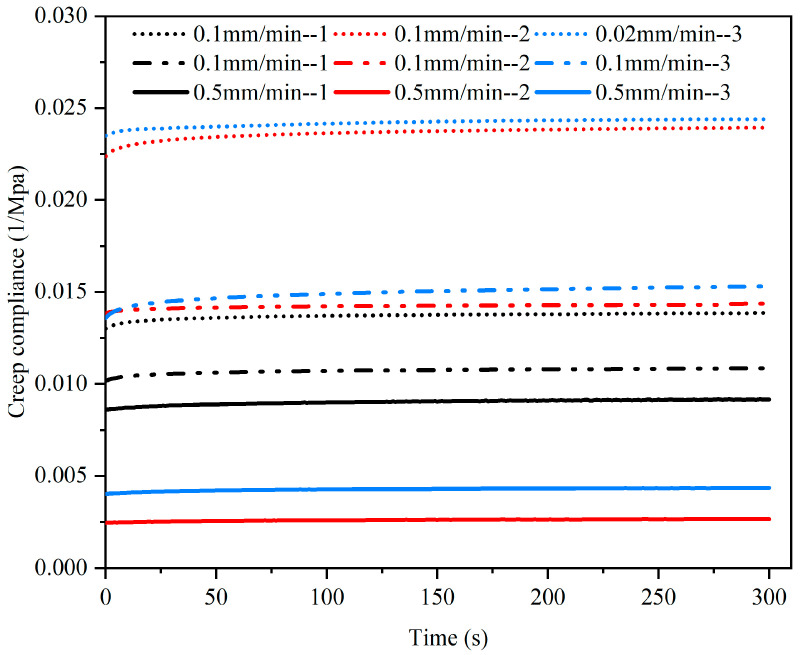
Creep compliance–time curves of three samples at different rates.

**Figure 12 materials-15-05912-f012:**
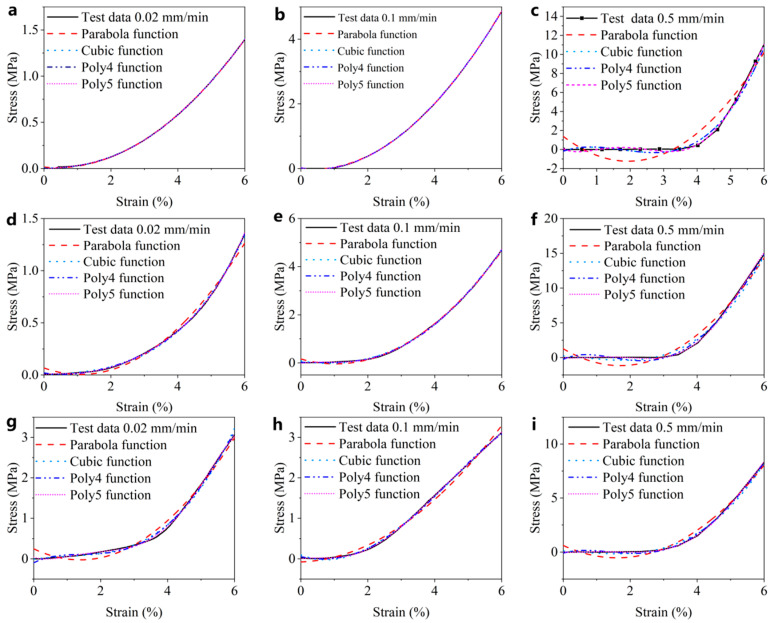
The fitting curves of four polynomial hyperelastic models. (**a**–**c**) Fitting curve of sample 1 at different rates; (**d**–**f**) fitting curve of sample 2 at different rates; (**g**–**i**) fitting curve of sample 3 at different rates.

**Figure 13 materials-15-05912-f013:**
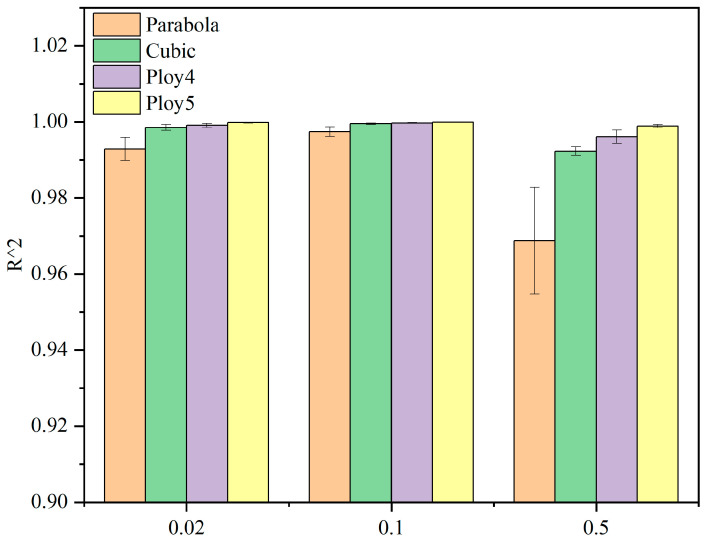
The fitting degree of four hyperelastic models.

**Figure 14 materials-15-05912-f014:**
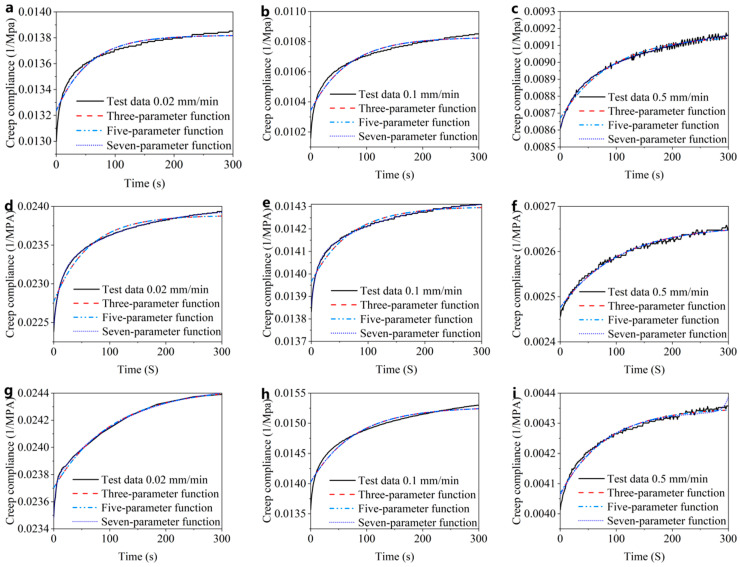
The fitting curves of three viscoelastic models. (**a**–**c**) Fitting curve of sample 1 at different rates; (**d**–**f**) fitting curve of sample 2 at different rates; (**g**–**i**) fitting curve of sample 3 at different rates.

**Figure 15 materials-15-05912-f015:**
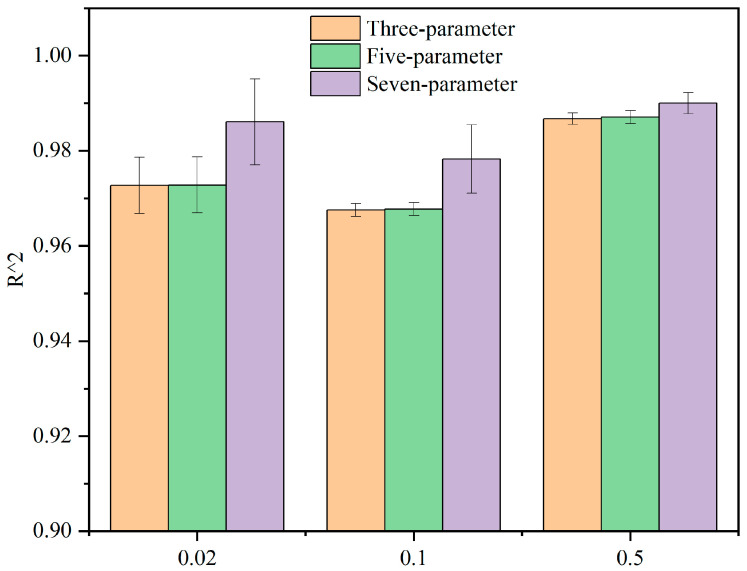
The fitting degree of the three viscoelastic models.

**Table 1 materials-15-05912-t001:** CBAB sample parameters.

Sample of CBAB	Sample 1	Sample 2	Sample 3
Thickness (mm)	1.45	1.44	1.45
compression area (mm^2^)	3.73	3.66	3.76

**Table 2 materials-15-05912-t002:** Parameters of uniaxial compression test.

Test	Loading Rate (mm/min)	Maximum Strain (%)
Uniaxial compression	0.02 0.1 0.5	6 6 6

**Table 3 materials-15-05912-t003:** Parameters of compression creep test.

Test	Loading Rate (mm/min)	Hold Load (N)	Hold Time (s)
Compression creep	0.02 0.1 0.5	50 50 50	300 300 300

## Data Availability

Not applicable.

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
