# Peer review of "Mechanical Behavior of Human Cancellous Bone in Alveolar Bone under Uniaxial Compression and Creep Tests"

_materials, 2022, doi:10.3390/ma15175912_

Round 1

Reviewer 1 Report

See attached file:

Reviewer 2 Report

Manuscript ID: materials-1835413

Title: Mechanical behavior of human cancellous bone in alveolar bone under uniaxial compression and creep tests

1.What is the main question addressed by the research?

To assess the compressive mechanical behavior of the human cancellous bone in alveolar bone (CBAB) with two experimental models (hyperelastic and viscoelastic).

2.Is it relevant and interesting?

The article is relevant and interesting.

3.How original is the topic?

The topic is current.

4.What does it add to the subject area compared with other published material?

The authors have collected and analyzed original data.

5.Is the paper well written?

Yes, the article is well written.

6.Is the text clear and easy to read?

Yes, but minor English editing is required.

7.Are the conclusions consistent with the evidence and arguments presented?

Yes, the conclusions consistent with the evidence and arguments presented but further studies are needed.

8.Do they address the main question posed?

Yes, the Authors addressed the main question posed.

Other comments:

·         English language: Minor spell check required

·         Summary of abbreviations required.

·         Introduction: This section needs few improvements. For example, Authors may include a brief sentence at the beginning of this section regarding innovations on this topic based on the following reference: <<Innovative materials and technologies to improve knowledge on mechanical behavior of human cancellous alveolar bone are an intense research topic in dentistry [https://doi.org/10.3390/jpm12010108]>>.

The other sections have been properly prepared.

After making the indicated changes, the article may be suitable for publication after Editorial evaluation.

Thanks for the opportunity to review this manuscript.

Reviewer 3 Report

This paper explains

In the process of orthodontic treatment, the remodeling of cancellous bone in alveolar bone (CBAB) is the key to promote the tooth movement. Study on the mechanical behavior of CBAB is helpful to predict the displacement of teeth and achieve the best effect of orthodontic treatment. To achieve this goal, the compressive mechanical behavior of the human CBAB was studied exper- imentally and a high accuracy constitutive model (hyperelastic and viscoelastic) has been built based on test results of the CBAB. Three CBAB samples were cut from the alveolar bone around the root apex of human teeth. Uniaxial compression test was used to study the transient elastic proper- ties of CBAB. Creep test was used to studied the time-dependent viscoelastic properties of CBAB. Both tests were carried out at the loading rates of 0.02 mm/min, 0.1 mm/min and 0.5 mm/min re- spectively. According to the results of uniaxial compression test and compression creep test, CBAB 23
had been proved to be a nonlinear viscoelastic and hyperelastic material. The stress-strain curve obtained from uniaxial compression test could be divided into three stages: the collapse stage of the front section, the exponential stage of the middle section and the almost linear stage of the rear end. According to the strain-time curve obtained from the compression creep test, the trend of strain increasing with time was relatively obvious in the first 30 s. After 200 s, the curve gradually tended to be flat. Four hyperelastic models and three viscoelastic models were used to fit the test data re-spectively. Finally, the fifth-order polynomial hyperelastic model (coefficient of determination R2>0.999) was used to describe the hyperelastic properties of CBAB, and the seven-parameter model of generalized Kelvin modified model (R2>0.98) was used to describe the viscoelastic properties of CBAB

The paper is good and well written. However the following concerns must be addressed

1.      For abbreviation like CBAB separate nomenclature must be defined for all such kinds of words. It will be then easy for readers.

2.      Sample preparation is discussed but its CAD modelling and fabrication is not explained in the paper. I will suggest to write it in the main file or supplementary

3.      Compression testing video need to be added for justification of experiments performed in real time or not. Add a video to supplementary file for readers

4.      How parameters in Table 1 are selected any reasons or logical explanation?? For example, 1.2 then why not 2 or some other values?

5.      Compare your results with literature for verifying your contribution to existing literature or your improvements in existing work along with reference in Table.

6.      Updated your old reference by replacing it with latest research papers

Round 2

Reviewer 3 Report

The paper is much improved and almost many changes are done as per suggestions and comments. However,  the following two points still need further improvement

1. The experimental testing video as a supplementary file is not provided. Even if it's not recorded you can do the experiment again and make a perfect video of test specimen preparation and testing.

2. The comparison of current work with literature is added to a revised version of the paper. However, the results are compared in different contexts. Please compare it with the same parameters. For example, you talked about human samples then comparison with literature will be with this parameter.
